# Radiographic Measurements of the Cardiac Silhouette and Comparison with Other Radiographic Landmarks in Wild Galahs (*Eolophus roseicapilla*)

**DOI:** 10.3390/ani11030587

**Published:** 2021-02-24

**Authors:** Petra Schnitzer, Shivananden Sawmy, Lorenzo Crosta

**Affiliations:** Avian, Reptile and Exotic Pet Hospital, Sydney School of Veterinary Science, The University of Sydney, 415 Werombi Road, Camden, NSW 2570, Australia; shivananden.sawmy@sydney.edu.au (S.S.); lorenzo.crosta@sydney.edu.au (L.C.)

**Keywords:** galah, *Eolophus roseicapilla*, radiographic imaging, cardiac silhouette, heart size, parrot, avian

## Abstract

**Simple Summary:**

Cardiac diseases are a common finding in captive parrots. In this retrospective study, the cardiac silhouette of 36 wild galahs was measured in the ventrodorsal and laterolateral projections using radiographic images. The aim of the study was to determine reference values of the width and the length of the heart of wild galahs in relation to other anatomic landmarks. The evaluated reference indicates that the width of the heart is 50–65% of the thoracic width and 570–743% of the coracoid width.

**Abstract:**

Background: Part of the diagnostic workup for cardiac diseases is radiographic imaging. To determine an enlarged heart, species-specific reference values are necessary. Wild birds are rarely diagnosed with cardiac disease, and only a few studies have been done to investigate the cardiac silhouette in wild birds. Methods: In this retrospective study, the cardiac silhouette of 36 wild galahs, presented at the hospital, was investigated in relation to other anatomic landmarks like the thoracic width, clavicula width, synsacrum width, distance between the third and fourth rib, distance of the clavicula, and length and height of the sternum using a digital DICOM viewer. Results: The cardiac width was significant compared to the thoracic width with a minimum to maximum of 50 to 65%. The cardiac width compared with the coracoid width also showed significant results with a minimum to maximum range of 570 to 743%. A significant correlation was found between the weight and the cardiac width and length. Conclusion: The cardiac silhouette in wild galahs is easily measured in both radiographic views, and the heart size can be compared to other anatomical landmarks.

## 1. Introduction

Radiography is a common diagnostic tool in avian medicine and part of the diagnostic process for cardiac diseases. Cardiac diseases, whether primary or secondary, are common in bird species in captivity. In post-mortem findings, Krautwald-Junghanns et al. reported finding macroscopic changes of the heart in 36% of the cases, with 15% of these cases being hypertrophic or dilatative cardiomyopathy [1,2]. Another study described 9.7% of pet birds with cardiomyopathies at post-mortem examination and estimated that 5.6% of these birds likely died as a result of primary cardiac disease [3]. Free-living birds, on the other hand, have rarely been diagnosed with cardiac failure [2,4].

To improve the accuracy of the diagnosis, many studies have attempted to establish reference values of the cardiac silhouette in different avian species, including African grey parrots (*Psittacus erithacus*), Senegal parrots (*Poicephalus senegalus*), orange-winged Amazon parrots (*Amazona amazonica*), blue-fronted Amazon parrots (*Amazona aestiva*), budgerigars (*Melopsittacus undulatus*), Spix’s macaws (*Cyanopsitta spixii*), peregrine falcons (*Falco peregrinus*), Harris’s hawks (*Parabuteo unicinctus*), saker falcons (*Falco cherrug*), lanner falcons (*Falco biarmicus*), red-tailed hawks (*Buteo jamaicensis*), bald eagles (*Haliaeetus leucocephalus*), ospreys (*Pandion haliaetus*), screech owls (*Otus asio*), Canada geese (*Branta canadensis*), common kestrels (*Falco tinnunculus*), Bonelli’s eagles (*Aquila fasciata*), and Humboldt penguins (*Spheniscus humboldti*) [4,5,6,7,8,9,10,11,12,13,14,15]. The correlations measured between the heart size and stable points in the body could be used to direct the practitioner and identify alterations in heart size. A correlation between the width of the cardiac silhouette and the width of the cranial coelom could be found in most of the studied species [5,6,9,10,14,15]; however, coracoid and heart width were only correlated in a few species [6,9,15].

In this retrospective study, the cardiac silhouettes of wild galahs (*Eolophus roseicapilla*) were investigated in radiographic images. The aim of this study was to measure the cardiac silhouette of wild galahs in relation to other anatomic landmarks as reference values for this species.

## 2. Materials and Methods

The radiographs of 36 adult wild galahs were used to measure the cardiac silhouette. Medical records from January 2015 to July 2020 were searched using the diagnostic imaging software Asteris Keystone. All birds were wild and were presented to the Avian Reptile and Exotic Pet Hospital at the University of Sydney for triage and treatment from public or wildlife rescue associations. As part of the birds’ diagnostic workup, the animals underwent radiographic imaging under general anaesthesia with isoflurane (Isothesia NXT, Henry Schein, 2–4% isoflurane, oxygen 2 L/min) for proper positioning [16]. None of the patients included in this study showed signs of subclinical cardiac disease according to the physical examination and showed excellent body condition [17]. Blood (PCV and TP), taken as a routine examination, showed normal values as compared with reference values [18]. All birds in this study were judged perfectly fit and could be released after their hospital check-up. The determination of fitness level included an evaluation of body condition, plumage, minimal bloodwork, and indoor fly trials and allow for a clinical assessment of respiratory and heart function [19].

For inclusion in this study, the radiographs had to meet the following criteria: superimposition of the sternum with the spine in the ventrodorsal (VD) projection and superimposition of the two coracoids in the laterolateral (LL) projection. Animals that presented traumatic injury that might have altered the anatomy and location of the inner organs as well as birds with infectious diseases or fractures of the shoulder girdle were excluded from the study. The survey digital radiographs were all taken on a commercial X-ray machine (Cuattro DR, Golden, CO, USA) using exposure factors of 70 kV and 2.5 mAs. Each measurement was taken after accurate calibration on a Digital Imaging and Communications in Medicine (DICOM) viewer (Asteris Keystone) using a left/right mark as standard.

Measurements: In the VD projection, the width of the heart (cardiac silhouette width, CW) was measured at the widest point using the system ruler on the Asteris Keystone in mm (Figure 1). The width of the cranial coelom (thorax width, TW) was measured at the same height as the width of the heart (Figure 1). The width of the coracoid bones (coracoid width, CoW) was measured directly under the scapulohumeral joints to the nearest mm. The distance between the third and fourth ribs (DR) was measured parallel to the spine. The width of the synsacrum (synsacrum width, SynW) and the distance between the clavicles (distance clavicle, DC) and from the cranial to scapulohumeral joints were measured as described by Geerinckx et al. [20] (Figure 1). In the LL projection, the length of the heart (cardiac silhouette length, CL) starting from the aorta to the cardiac apex in addition to the length (sternum length in lateral, SL) and height of the sternum (sternum height, SH) were measured. The insertion of the coracoid to the sternum was used as a landmark to measure the length of the sternum and at 90° the height to the nearest mm (Figure 2). Data for bird weight were incomplete; in 12 out of 36 birds, no weight could be evaluated, precluding statistical comparison with the heart size in only 24 birds.

Data analysis: The statistical analysis was performed using a commercially available statistics program (SPSS, IBM^®^, Armonk, NY, USA). The Shapiro–Wilk test was performed to investigate normality and the Pearson correlation coefficient was used to test for correlation. Homoscedasticity and linearity were evaluated graphically. The heart width and its relationship with the widths of the cranial coelom, coracoid, and synsacrum as well as the distance between clavicles and between the third and fourth rib were evaluated by linear regression. On the LL projection, the relationship between the length of the heart and the length and depth of the sternum was also evaluated by linear regression. The confidence interval was set at 90% due to the small sample size. The cardiac width and length were considered dependent variables, and each radiographic index was an independent variable. The minimum, maximum, median, mean, and standard deviation were also calculated and expressed as mean ± SD. Significance was considered with values *p* < 0.05.

## 3. Results

The results show that the ratio of the cardiac silhouette is 50–65% of the width of the cranial coelom. The width of the cardiac silhouette is 570–743% of the coracoid width, as measured caudally to the shoulder joint. The distance between clavicles is 83–124% of the cardiac width. The mean values ± SD, minimum, maximum, median, and confidence intervals are listed in Table 1 and Table 2. All variables were normally distributed, except for the clavicle measurement, which included three outliers.

A significant weak positive correlation was found between the CW and TW (R^2^ = 0.196; *p* = 0.003) with a regression formula as follows:CW (mm) = 10.755 + (TW (mm) ∗ 0.288).

Between CW and CoW (R^2^ = 0.133; *p* = 0.029), a significant weak positive correlation was also found with a regression formula as follows:CW (mm) = 13.778 + (CoW (mm) ∗ 2.309).

The correlation between CW and TW and CW and CoW is demonstrated in Figure 3 and Figure 4 in scatter plots.

A significant correlation is also present between the heart size and the body weight. In 24 out of 36 birds, the body weight was reported over the course of the physical examination. The significance and correlation are reported as follows: body weight and CW, r = 0.459, *p* = 0.012; body weight and CL, r = 0.452, *p* = 0.007. The mean body weight in the group was 275 ± 52.43 g. No other significant correlations between variables could be identified (CW and DR, r = 0.088, *p* = 0.306; CW and SynW, r = 0.098, *p* = 0.285; CW and DC, r = 0.129, *p* = 0.227). On the LL projection, the length of the heart was between 43% and 70% of the sternum length. Although not significant, there was a trend for the length of the heart to increase with the length (r = 0.232, *p* = 0.087) and height (r = 0.211, *p* = 0.108) of the sternum.

## 4. Discussion

The results of the present study show that the cardiac silhouette of wild galahs in the VD projection is 21.38 ± 1.34 mm wide, 50–65% of the width of the cranial coelom, and 570–743% of the width of the coracoid. The cardiac silhouette in the LL projection is 27.60 ± 3.12 mm long and between 43% and 70% of the sternum length. During the study, it was easy to measure the width of the heart in the VD projection as well the length of the heart in the LL view. Unfortunately, the length of the heart in the VD projection and the width of the heart in the LL projections could not be measured in all images due to superimposition of the liver and proventriculus. The decision to measure the other anatomical structures (TW, CoW, DC, DR, SynW, SL, and SH) was based on other studies measuring the cardiac silhouette. In other studies, the heart length was not measured. Among all Psittaciformes, only in cockatoos is the apex of the heart clearly visible, and no superimposition of the liver is present [20,21]. Like in other species of Psittaciformes, the heart in cockatoos is superimposed by the proventriculus and, thus, its height and other parameters cannot be measured precisely in the LL projection [20], which was also the case in our study.

The main indication for performing radiography in the wild galahs presented to our hospital was trauma. Birds with fractures of the shoulder girdle, inner coelomic lesions, or any other traumatic or pathologic lesions that could have altered the measurements in the radiography image were excluded from the study. A total number of 27 birds were excluded from the study, due to severe lesions or disease.

In addition, it was not possible to calibrate the DICOM viewer in three cases, and this resulted in the exclusion of these patients from the study.

One of the selection criteria was the superimposition of the sternum with the spine in the VD projection plus superimposition of the insertion point of the coracoid in the sternum. In cases where the carina was distinctly separated from the spine (VD view), the animal was excluded from the study. These indications resulted in the exclusion of 11 animals, which limited this study.

In the present study, the relation between weight and heart size showed a moderate significant correlation with r = 0.459 and *p* = 0.012. The sex/weight ratio does not appear statistically relevant in budgerigars according to one study [14], but it was suggested to be relevant in other species [5]. In our study, a correlation between the weight and the cardiac width and length could be verified. The sex of our birds was not known as determination of sex is not part of the common procedure in our hospital. Further studies are necessary to identify a possible correlation between the sex and the heart width and length.

The results of the present study suggest that in wild galahs, the width of the cardiac silhouette is approximately 50–65% of the width of the cranial coelom and 570–743% of the width of the coracoid. In studies performed in other species, similar values were measured, although our study showed the highest maximum range with 65%. Straub et al. [6] described the heart width to be within 51–61% of the cranial coelom width and 545–672% of the width of the coracoid in captive African grey parrots, Senegal parrots, and orange-winged Amazon parrots. In captive blue-fronted Amazon parrots, the cardiac width compared to the cranial coelomic width ranged between 40% and 45%.

In budgerigars, the cardiac width is reported to be the 62% of the thoracic width [14], and in Spix’s macaws it ranges from 46 to 60% [10].

In non-parrot species, the cardiac/thoracic width ratio showed considerable variation. It is reported to be 45–59% in Humboldt penguins [8], 66–72% in saker falcons [9], 54–61% in Harris’s hawks [9], 66–74% in peregrine falcons [9], 65–72% in lanner falcons [9], 48–57% in Bonelli’s eagle [15], 67–69% in ospreys, 44–52% in bald eagles and 51–75% in common kestrels [4].

In these previous studies, only the common kestrels and the ospreys were wild animals and showed the widest ratio, with a maximum value of 75% [4] of the thoracic width. Migrating birds and birds performing exercise have a larger heart than non-exercising birds [17,22]. Beaufrère and Fitzgerald describe the larger hearts of wild cockatoos compared with captive species as sign of their cardiac fitness [19]. The falcon species studied in previous studies showed generally larger cardiac width than parrot species [6,9,13]. In Mirshahi et al. [4] the width of the heart in wild common kestrels was larger compared to other falcon species, and in our study, our measured heart width was larger compared to that in other parrot species. Further investigations are necessary for a similar comparison of cardiac silhouette in captive galahs.

However, in Pees et al. [2], the heart size of wild galahs (rose-breasted cockatoos—*Eolophus roseicapilla*) was investigated, after euthanasia, to evaluate anatomy and pathology. In the study, the heart width was determined to be 17.7–27.3 mm (20 ± 1.8), the heart length 19.5–30.3 mm (26 ± 2.2), and the heart height 14.1–18 mm (16 ± 0.9) [2]. In the present study, the width was within the same range presented by Pees et al., 17.77–23.63 mm (21.38 ± 1.34). The heart length in our study was longer at 22.73–37.15 mm (27.60 ± 3.12). The difference in the length is possibly due to differences in the method and landmarks used for measurements in the two studies; in our study, we measured the heart from a radiographic image, while Pees et al. measured the heart following explant from the cadaver.

Cardiac diseases with different aetiology are common in captive birds, but still it is not easy to diagnose heart problems in living patients [19,23,24,25,26,27].

Krautwald-Junghanns (2004) [1] reports that 15% of 36% birds with macroscopic changes of the heart had a hypertrophic or dilatative cardiomyopathy. Furthermore, the same paper reports that 6% of 36% of birds with macroscopic cardiac changes had a pericardial effusion. It is reported that 21% of birds with a heart problem have a cardiac disease that leads to a cardiomegaly and, hence, this could be observed with the application of radiography. Further studies are necessary to obtain reference values for possible application in diagnosis.

## 5. Conclusions

This report is the first to measure the cardiac silhouette of wild galahs and of wild Psittaciformes, in general, using radiographs and providing so reference values for wild galahs. These findings indicate that cardiac measurements are mostly similar between the birds but that differences do exist within the species.

About 21% of birds with a cardiac issue have a heart disease that will cause cardiomegaly and, therefore, can be identified with radiographs [1].

Because of this, species-specific measurements are necessary to help guide the clinician when using radiographs to assess cardiac size and plan for cardiac workup and, hence, signify the importance of studies that establish reference values for wild and captive birds, which can also help to understand the causes of cardiac diseases that develop in birds in captivity but are almost non-existent in the wild [2].

## Figures and Tables

**Figure 1 animals-11-00587-f001:**
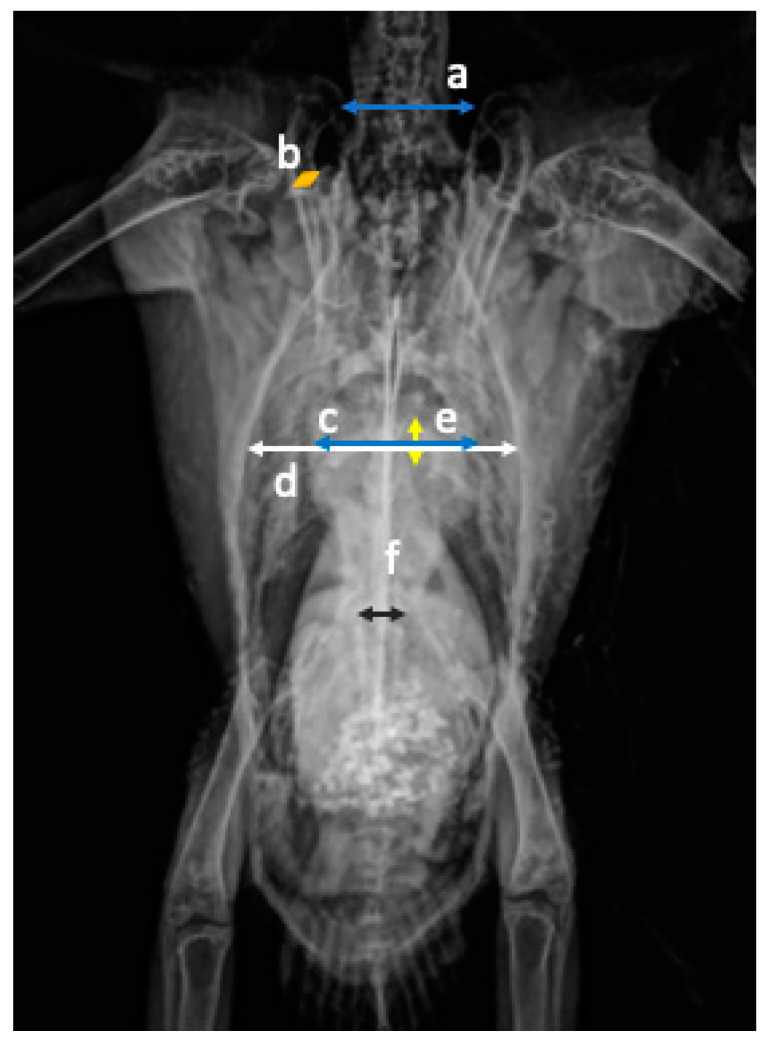
Ventrodorsal (VD) projection and measurements in a radiograph of a wild galah: (**a**) distance of the clavicle cranial to the shoulder joint (blue); (**b**) coracoid width immediately caudal to the shoulder joint (orange); (**c**) width of the heart at the widest point (light blue); (**d**) width of the thorax at the same height as the heart width (white); (**e**) distance between the third and fourth ribs parallel to the spine (yellow); (**f**) synsacrum at the widest point (black).

**Figure 2 animals-11-00587-f002:**
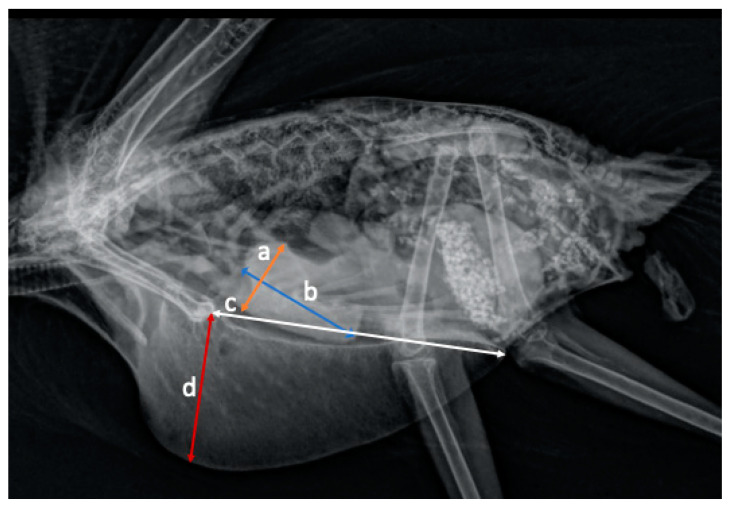
Laterolateral (LL) projection and measurements in a radiograph of a wild galah: (**a**) widest point of the heart (orange); (**b**) length of the heart at height of the aorta and pulmonary artery to the apex (light blue); (**c**) length of the sternum from the insertion of the coracoid to the caudal edge (white); (**d**) depth of the sternum, 90° from the junction of the coracoid to the sternum (red).

**Figure 3 animals-11-00587-f003:**
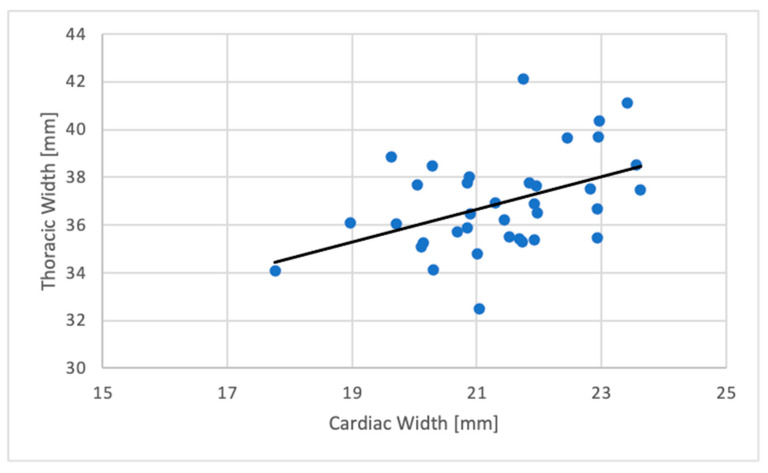
Scatterplot showing the relation between the cardiac width (mm) and the cranial coelom width (mm) in the VD radiographic projection (*N* = 36).

**Figure 4 animals-11-00587-f004:**
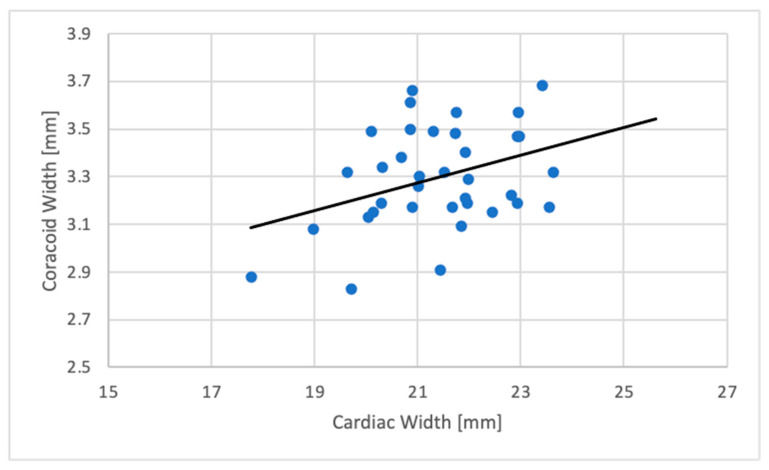
Scatterplot showing the relation between the cardiac silhouette width (mm) and the coracoid width (mm) in the VD radiographic projection (*N* = 36).

**Table 1 animals-11-00587-t001:** Measurements from the VD radiographic projection. The mean value with standard deviation, minimum, maximum, and confidence intervals of the variables are reported.

		Mean ± SD	Median	Minimum	Maximum	90% Confidence Interval
Cardiac width (mm)	CW (*N* = 36)	21.9 ± 1.3	21.5	17.8	23.6	21.0–21.8
Thoracic width (mm)	TW (*N* = 36)	36.9 ± 2.1	36.6	32.5	42.1	36.3–37.5
Coracoid width (mm)	CoW (*N* = 36)	3.3 ± 0.2	3.3	2.8	3.7	3.2–3.4
Distance between third and fourth rib (mm)	DR (*N* = 36)	6.2 ± 0.6	6.3	5.00	7.3	6.0–6.4
Synsacrum width (mm)	SynW (*N* = 36)	5.4 ± 0.4	5.4	4.4	6	5.3–5.5
Distance clavicle (mm)	DC (*N* = 33)	20.6 ± 1.9	20.4	16.4	27.6	20.1–21.1

**Table 2 animals-11-00587-t002:** Measurements from the LL radiographic projection. The mean value with standard deviation, minimum, maximum, and confidence intervals of the variables are reported.

		Mean ± SD	Median	Minimum	Maximum	90% Confidence Interval
Cardiac length (mm)	CL (*N* = 36)	27.6 ± 3.1	27.1	22.7	37.2	26.7–28.5
Sternum length (mm)	SL (*N* = 36)	51.1 ± 2.3	51.0	46.7	56.3	50.4–51.7
Sternum height (mm)	SH (*N* = 36)	23.5 ± 2.6	24.1	15.9	27.9	22.8–24.2

## Data Availability

Database University of Sydney.

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
