# Peer review of "Radiographic Measurements of the Cardiac Silhouette and Comparison with Other Radiographic Landmarks in Wild Galahs (Eolophus roseicapilla)"

_animals, 2021, doi:10.3390/ani11030587_

Round 1
Reviewer 1 Report
Line 18: The term cardiomyopathies is not appropriate. Cardiac disease would be more appropriate. CM refers to disorders of the myocardium only. We see pericardial effusion, etc as well, which is diagnosed by radiographs initially.
Line 24: remove name of dicom viewer from abstract.
Line 25 : what do these values (e.g. 50-65%) mean? Are they ranges? Reference intervals? Remove the p values, unclear what they refer to.
Line 28: “wild bird species seem to have larger hearts” is not a conclusion you can draw from your study, since you only looked at one species and you did not compare wild galahs to captive galahs. Remove.
Line 34: “cardiomyopathies” replace with “cardiac disease” correct throughout manuscript.
Line 52: remove the term “cross-sectional study”.
Line 54: your hypothesis is flawed and cannot be tested. Any species captive or wild can be used to calculate reference intervals. you cannot rule-out that the animals used in your study did not have subclinical cardiac disease nor can you assess if your reference values are “accurate”.
Line 64: all animals had echocardiography or blood work performed? What blood tests are you referring to, which can rule out cardiac disease. I am not aware of any sensitive blood marker to assess cardiac disease.
Line 106: not sure what relationship besides correlation you are trying to assess and how this is necessary based on your hypothesis? All linear regression information should be removed from the manuscript and correlation reported for the different measurements.
Line 113-14: 50-60% and 570-743% ,etc is the range, the reference interval? Unclear.
Line 117: do not report homoscedasticity. Not relevant.
Line 120 and 124: a weak correlation is clinically irrelevant. Do not report the regression formulas. There is no clinically relevant correlation between these parameters.
Line 140: replace “is” with “was” correct throughout manuscript. It should be written in past tense, not present tense.
Table 1 and 2: report only one decimal point for all values, not two.
Author Response
Line 18: The term cardiomyopathies is not appropriate. Cardiac disease would be more appropriate. CM refers to disorders of the myocardium only. We see pericardial effusion, etc as well, which is diagnosed by radiographs initially.
The authors agree, this was a mistake and has been changed.
Line 24: remove name of dicom viewer from abstract.
The name of the dicom viewer has been removed from the abstract. It will be named in materials and methods.
Line 25 : what do these values (e.g. 50-65%) mean? Are they ranges? Reference intervals? Remove the p values, unclear what they refer to.
The values are the width of the heart in relation to the width of the thorax The same with the width of the heart in relation to the width of the coracoid. The relation between the width of the heart and the width of the thorax respectively the width of the coracoid were the only two measurements with a statistical significance in this study.
The choice to present the results as percentage derives from other papers published prior to our study. In 6 out of 11 papers the values have been presented as percentage. In 3 out of 11 as a ratio and in another 3 papers as absolute numbers in mm. For that reason, we decided to present the results as a percent.
I rephrased the sentence to make it clearer.
Line 28: “wild bird species seem to have larger hearts” is not a conclusion you can draw from your study, since you only looked at one species and you did not compare wild galahs to captive galahs. Remove.
Removed and placed only in the discussion, as it is a citation from Cardiology Chapter of Speer’s Current Therapy in Avian Medicine and Surgery (2016), page 253.
Line 34: “cardiomyopathies” replace with “cardiac disease” correct throughout manuscript.
Line 41, 189: The word cardiomyopathies is a direct citation from the cited literature.
Line 11, 18, 38: Cardiomyopathies changed to cardiac diseases.
Line 52: remove the term “cross-sectional study”.
The term cross sectional studies have been removed.
Line 54: your hypothesis is flawed and cannot be tested. Any species captive or wild can be used to calculate reference intervals. you cannot rule-out that the animals used in your study did not have subclinical cardiac disease nor can you assess if your reference values are “accurate”.
Line 59: changed to limit the hypothesis to the observation of the cardiac silhouette with the anatomical landmarks. In the cardiology chapter of Speer’s Current Therapy in Avian Medicine and Surgery, page 253, the authors make the connection of cardiac fitness in a study performed on wild Cockatoos (see article below) with a relative greater thickness of the left ventricle compare to the thickness in captive budgerigars and Australian king parrots.
- Pees, M.; Zeh, C.; Filippich, L.J.; Krautwald-Junghanns, M.-E. Pathologisch-anatomische und morphometrische Untersuchungen am Herzen von wildlebenden Kakadus. Tierärztl. Prax. Ausg. K Kleintiere Heimtiere 2014, 42, 390–396, doi:10.15654/TPK-131101.
Summary
Objective: To evaluate the heart of free-living psittacine birds macroscopically and morphologically, and to compare the results to findings published for psittacine birds living in captivity to obtain information on the influence of bird keeping in a human environment on the psittacine heart.
Material and methods: In total, 84 wild-living cockatoos were examined, including 50 sulphur-crested cockatoos (Cacatua galerita), 31 galahs (Eolophus roseicapilla) and three long-billed corellas (Cacatua tenuirostris). The birds were euthanized because of a local cockatoo control program in Australia, and were examined pathologically within 8 hours of euthanasia. A macroscopic necropsy was performed, and the heart was assessed morphologically. Furthermore, a histological organ screening was conducted.
Results: The birds demonstrated good body condition and excellent muscle condition. Except for some paleness of the heart muscle, none of the animals showed any pathological alteration of the heart or large vessels. The mean heart mass was 8.7 g for the sulphur-crested cockatoos, 5.3 g for the galahs and 8.6 g for the long-billed corellas. Independent of the species examined, a highly significant correlation was found between the heart and body masses (r = 0.91; p < 0.001), which was also confirmed as significant within the sulphur-crested cockatoo (r = 0.59; p < 0.001) and galah groups (r = 0.52; p = 0.003). This correlation can be used to calculate the expected heart mass based on the body mass, using the formula: heart mass (g) = 2.9 + 0.01 × body mass (g). In comparison to reports on Australian parakeets, the relative thickness of the heart muscle wall of the left ventricle found in this study was greater.
Conclusion: In comparison to psittacine birds kept in captivity, wild-living cockatoos have good body condition and rarely suffer from macroscopically detectable diseases of the heart or large vessels. The cardiac fitness level is superior in comparison to that found in healthy appearing psittacine birds kept in captivity.
Clinical relevance: The results can serve as a basis for the assessment of the heart in psittacine birds, because in contrast to earlier reports, the heart of healthy psittacine birds not previously exposed to any human influence could be assessed.
Line 64: all animals had echocardiography or blood work performed? What blood tests are you referring to, which can rule out cardiac disease. I am not aware of any sensitive blood marker to assess cardiac disease.
Line 69-72: rephrased as not clear. The blood work performed is routinely done with PCV and TP as a quick check up for severe blood loss. Ultrasound was performed, but the purpose was to collect data for another study. As it has been started recently, not all the animals in this study has been scanned with ultrasounds. For that reason, the authors decided to remove the ultrasounds from the article. I agree, that there are no sensitive blood markers in birds, but severe hypovolemia and hypoproteinemia alter the heart function secondarily, therefore the assessment of PCV and TP was considered of some value
Line 106: not sure what relationship besides correlation you are trying to assess and how this is necessary based on your hypothesis? All linear regression information should be removed from the manuscript and correlation reported for the different measurements.
Dear reviewer,
Thank you for the comment. The correlation has been done with all the measurements.. We used the linear regression because all the previous papers treating similar topics in birds, used the linear regression as a statistic tool. The statistical data have been revised by a teacher of statistics at the Louisiana State University and he suggested to present the data in this way. We will be happy to change the information, if you can kindly explain to us, the reason why you would like the data to be reported in a different way.
Line 113-14: 50-60% and 570-743%, etc is the range, the reference interval? Unclear.
Line 123, 161: as many other articles refer to percentages, the authors choose the same method to allow readers to compare easily between the species.
Line 117: do not report homoscedasticity. Not relevant.
Removed.
Line 120 and 124: a weak correlation is clinically irrelevant. Do not report the regression formulas. There is no clinically relevant correlation between these parameters.
Line 147-153: we reported correlations as they resulted. We can agree there is low value, clinically, but in our opinion, the numbers have to be reported, at least in the discussion. .
Line 140: replace “is” with “was” correct throughout manuscript. It should be written in past tense, not present tense.
The article will be send to the editing service, as none of the authors is native English speaker.
Table 1 and 2: report only one decimal point for all values, not two.
Changed as requested.
Dear reviewer,
Thank you for the extensive revision. I am acting as reviewer for several Journals, as well and I know how much work it takes.
You mentioned that the results are not clearly presented, the conclusions are not supported by the results and that there is not enough background.
We hope in this revised version the data are presented in better way. We strongly believe our conclusions are supported by the results and if on one side the current literature lacks information about avian cardiology, on the other hand the few available papers seem to support out hypothesis. Not only, we believe that the fact that there are few publications about avian cardiology, compared to cardiology in other species, should be a motivation to support studies and publications about this interesting field.
I hope I have added enough information and literature to give background and that the conclusions are now clearer. It is not clear to me how to improve the presentation of the results. I tried to follow the guidelines for other articles and our statistician.
I hope it is overall now improved.
Kind regards,
The authors
Reviewer 2 Report
Dear Authors,
Many thanks for producing data on the use of conventional radiology in measuring the cardiac silhouette in in wild Galahs. This is an interesting study and the authors have collected a unique dataset. The paper is generally well written and structured. However, in my opinion, although measurements may have value in the descriptive part of a radiology report, they should not be emphasized as a basis for diagnosis in either teaching or clinical imaging reports.Furthermore, emphasis on measurements is unwarranted when the pathologic effects of disease are invariably multiple and all the imaging signs must be recognized for optimal interpretation.Lastly it is important not to put too much emphasis on the radiographic appearance of the cardiac silhouette.
In order to increase the sensitivity of the test would be ideal to insert some patients with cardiac disease and to compare the radiological evaluation with echocardiography ( you did mention in your routine workout)because is the standard method for non-invasive assessment of cardiac dimensions, and able to discriminate the boundaries of the cardiac chambers from the neighbouring pulmonary veins.
Author Response
Dear reviewer,
Thank you for your comment. The ultrasound examination done was not performed in all animals. I changed it now as it was not clearly described what has been done. See line 70-73. The ultrasound examination is part of another study together with ultrasound in patients with cardiac disease. These are studies on captive parrots as well. For that reason the authors decided to split the research.
In this study we will limit to present the data measured as done in other similar studies. It is the first study of this type in wild parrots and so in free-flying birds. I hope that further studies together with the present study will give a deeper knowledge in diagnostic imaging of the heart in wild birds compared to caged birds.
kind regards,
the authors